# Combined Immunoinformatics to Design and Evaluate a Multi-Epitope Vaccine Candidate against *Streptococcus suis* Infection

**DOI:** 10.3390/vaccines12020137

**Published:** 2024-01-29

**Authors:** Song Liang, Shidan Zhang, Yinli Bao, Yumin Zhang, Xinyi Liu, Huochun Yao, Guangjin Liu

**Affiliations:** 1College of Veterinary Medicine, Nanjing Agricultural University, Nanjing 210095, China; 2OIE Reference Lab for Swine Streptococcosis, College of Veterinary Medicine, Nanjing Agricultural University, Nanjing 210095, China; 3Joint International Research Laboratory of Animal Health and Food Safety, College of Veterinary Medicine, Nanjing Agricultural University, Nanjing 210095, China; 4Key Laboratory of Animal Bacteriology, Ministry of Agriculture, College of Veterinary Medicine, Nanjing Agricultural University, Nanjing 210095, China; 5Engineering Research Center for the Prevention and Control of Animal Original Zoonosis, Fujian Province University, College of Life Science, Longyan University, Longyan 364012, China; 6Sanya Institute of Nanjing Agricultural University, Nanjing Agricultural University, Sanya 572000, China

**Keywords:** multi-epitope vaccine, *Streptococcus suis*, conformational epitope, immunoinformatic, immune protection

## Abstract

*Streptococcus suis* (*S. suis*) is a zoonotic pathogen with multiple serotypes, and thus, multivalent vaccines generating cross-protection against *S. suis* infections are urgently needed to improve animal welfare and reduce antibiotic abuse. In this study, we established a systematic and comprehensive epitope prediction pipeline based on immunoinformatics. Ten candidate epitopes were ultimately selected for building the multi-epitope vaccine (MVSS) against *S. suis* infections. The ten epitopes of MVSS were all derived from highly conserved, immunogenic, and virulence-associated surface proteins in *S. suis*. In silico analyses revealed that MVSS was structurally stable and affixed with immune receptors, indicating that it would likely trigger strong immunological reactions in the host. Furthermore, mice models demonstrated that MVSS elicited high titer antibodies and diminished damages in *S. suis* serotype 2 and Chz infection, significantly reduced sequelae, induced cytokine transcription, and decreased organ bacterial burdens after triple vaccination. Meanwhile, anti-rMVSS serum inhibited five important *S. suis* serotypes in vitro, exerted beneficial protective effects against *S. suis* infections and significantly reduced histopathological damage in mice. Given the above, it is possible to develop MVSS as a universal subunit vaccine against multiple serotypes of *S. suis* infections.

## 1. Introduction

*Streptococcus suis* (*S. suis*) is an important zoonotic pathogen that may cause systemic infections such as Streptococcal toxic shock-like syndrome (STSLS), meningitis, septicemia, and arthritis in millions of people with occupational contact with pigs and pork (e.g., farmers, slaughterhouse workers, and butchers) [1,2]. *S. suis* can infect the respiratory tract, the digestive tract and wounds [3,4]. The first human case of *S. suis* infection was reported in Denmark in 1968 [5], followed by more than 30 other countries in Europe, North America, South America, Oceania, and Asia [6,7]. *S. suis* serotypes are numerous and can be categorized into 29 classic serotypes (1–19, 21, 23–25, 27–31, and 1/2) and 21 novel cps loci (NCL1-20 and Chz) based on capsular polysaccharide (CPS) antigenicity [8]. *S. suis* serotype 2 (SS2) is the predominant epidemic type in clinical cases of humans worldwide [9]. In China, two outbreaks of SS2 in humans occurred in 1998 and 2005, resulting in 14 and 38 deaths, respectively [10,11]. In addition, *S. suis* serotypes 5 (SS5), 7 (SS7), and 9 (SS9) are all acknowledged to be critical potentially infectious serotypes to humans [12,13]. *S. suis* serotype Chz (SSChz) has been successively isolated in China [14] and Canada [15], showing SSChz has a global prevalence. An SSChz strain, CZ130302, isolated in China reproduced meningitis symptoms in mice models with a higher virulence than the highly virulent reference SS2 strain P1/7 [14]. 

*S. suis* has traditionally been prevented and treated with antibiotics [13]. However, the increasing resistance of *S. suis* poses a challenge to clinical medication [16]. Vaccination is an alternative means of preventing *S. suis* infections; however, existing vaccines fail to convincingly reduce the burden of highly prevalent serotypes of *S. suis* [17,18]. In the past decade, muramidase-released protein (MRP) [19], enolase [20], extracellular protein factor (EF) [19,21], and Sao [22] of *S. suis* have been identified as immunogenic proteins, and subunit vaccines developed from these proteins provide varying degrees on protection against *S. suis*. However, on the one hand, the pathogenesis of *S. suis* is complex, and traditional subunit vaccines fail to provide sufficient protection [18]. On the other hand, the various serotypes of *S. suis* demand highly conserved proteins to hopefully provide cross-protection [23,24]. Recently, enolase and Sao co-expression formed a candidate vaccine that provided cross-protection against *S. suis* in mice models [25]. In addition, the protein JointS, a combination of MRP, glyceraldehyde 3-phosphate dehydrogenase (GAPDH), and the novel dihydrolipoamide dehydrogenase DLD, provided favorable protection in a virulent SS2-infected piglet model [3], which presents an attractive approach: compared to single antigen vaccines, co-expression of multiple antigens provides better protection in livestock vaccines.

Identification of conserved and immunogenic surface proteins within a species has been made feasible through the establishment of reverse vaccinology and immunoinformatics [26]. Further, the minimal immunogenic region-antigenic epitope for candidate subunit proteins can be predicted [27]. Multiple candidate antigenic epitopes joined by linkers to target diverse conserved surface proteins has resulted in cross-protection and has been demonstrated to be a well-established and safety-stable strategy in several bacteria such as *Staphylococcus aureus* [28], *Escherichia coli* [29] and *Streptococcus pneumoniae* [30]. Recently, a universal multi-epitope candidate vaccine was constructed against *S. suis* infection in swine using an immunoinformatics approach [31]. However, the limited surface proteins of *S. suis* and the absence of experimental verifications appear to be insufficient to convincingly validate the availability of the vaccine. Therefore, the current protection data for multi-epitope vaccines against *S. suis* needs to be expanded. 

In this study, we used reverse vaccinology, immunoinformatics, and pan-genomics to analyze the currently completely sequenced *S. suis* (n = 120), selecting highly conserved proteins and screening the optimal candidate epitopes through a cascade of steps: protective antigen prediction, antigenicity, subcellular localization, B-cell epitopes (conformational and linear), and T-cell epitopes (linear). Finally, the candidate epitopes were linked and named MVSS, and the workflow is shown in Figure 1. Further, in vitro bacterial inhibition assays and in vivo mice models demonstrated that MVSS had positive cross-protective effects against *S. suis*. 

## 2. Materials and Methods

The workflow of integrated in silico analysis and mice challenge for developing a multi-epitope vaccine (MVSS) for *S. suis* is shown in Figure 1.

### 2.1. Pan-Genomic Analysis

To ensure that candidate proteins were conserved, all completely sequenced *S. suis* genomes (n = 120) were downloaded from NCBI. Prokka (version 1.14) was used to annotate the coding DNA sequences (CDSs) of every genome [32], and corresponding GFF3 format files were generated for uploading to Roary (version 3.13.0) [33]. 17,730 proteomes were analyzed, and genes found in 95% to 100% of genomes were considered core and soft-core genes.

### 2.2. Antigenicity Prediction

To screen for protective antigens, conserved protein sequences found in *S. suis* were submitted to VaxiJen (version 2.0) [34] (http://www.ddg-pharmfac.net/vaxijen/VaxiJen/VaxiJen.html) (22 September 2022) for predicting protective antigens, with proteins scoring ≥95% used as a subsequent screening. 

### 2.3. Subcellular Localization

Surface proteins and secreted proteins were deemed potential targets for subunit vaccines [23,24]. Therefore, previously screened protective antigens were uploaded to the PSORTb server (version 3.0.2; http://www.psort.org/psortb/) (23 September 2022) for subcellular localization prediction [35]. “Gram-positive” was selected, and other settings were set to defaults. Only the proteins predicted to be located in the cytoplasmic membrane, cell wall, and extracellular were selected for subsequent analysis.

### 2.4. T-Cell Epitope Screening

Adaptive immunity is mediated by B cells and T cells recognizing antigenic epitopes, where T cells recognize linear epitopes bound to major histocompatibility complex (MHC) molecules to bridge the cellular immune response [36,37]. In order to anticipate the major histocompatibility complex class II (MHC-II) binding epitopes, candidate protein sequences were submitted to the Immune Epitope Database website (IEDB; http://www.iedb.org/) (27 September 2022) [38]. Human-derived MHC II HLAs molecules (DRB1 * 01: 01, DRB1 * 03: 01, DRB1 * 04: 01, DRB1 * 07: 01, DRB1 * 11: 01, DRB1 * 13: 01 and DRB1 * 15: 01) were set to at least 50% of allele binding. In addition, binding predictions were performed for the entire HLA reference set using the IEDB-recommended 2.22 prediction method. 

### 2.5. B-Cell Epitope Screening

Different from T cells, B-cell epitopes are classified as conformational and linear epitopes regarding spatial structure [39,40]. The B-cell linear epitopes were predicted by the ABCPred online server (http://crdd.osdd.net/raghava/abcpred/) (1 October 2022) [41]. The screening threshold was set to 0.4. To filter for B-cell conformational epitopes, candidate proteins were initially modeled by Alphafold2 (version 2.0) (https://colab.research.google.com/github/sokrypton/ColabFold/blob/main/AlphaFold2.ipynb) (5 October 2022) (Appendix A), and the generated PDB format was uploaded to IEDB (DiscoTope: Structure-based Antibody Prediction) (http://tools.iedb.org/discotope/) (13 October 2022) for conformational epitope prediction [42]. Peptides with scores greater than −3.7 were considered promising.

B-cell and T-cell epitopes were aligned using local BlastN to select the peptides containing both B-cell and T-cell epitopes. Finally, antigenicity score >0.9 for developing vaccine (Table 1) was evaluated by VaxiJen.

### 2.6. Multiepitope Subunit Vaccine Design

To increase the solubility of the multi-epitope vaccine, selected antigenic epitopes were uploaded to ExPasy (version 3.0) (https://web.expasy.org/protparam/) (16 October 2022) for predicting hydrophilicity [43]. The antigenic epitopes were joined by GPGPG and LRMKLPKS to form MVSS consistent with the principle of increasing hydrophilicity from the center to the edges (Appendix A). To predict the antigenicity and sensitization of MVSS, VaxiJen, and Algpred (https://webs.iiitd.edu.in/raghava/algpred/submission.html) (16 October 2022) were used.

The three-dimensional (3D) model of MVSS was modeled by I-TASSER (https://zhanglab.ccmb.med.umich.edu/I-TASSER/) (17 October 2022) [44]. Furthermore, the models with high C scores were uploaded to the GalaxyRefine server (https://galaxy.seoklab.org/) (20 October 2022) for refinement [45]. Ramachandran plot analysis (saves.mbi.ucla.edu/) was used to evaluate the refined 3D model [46].

### 2.7. Molecular Docking and Molecular Dynamics Analysis

To determine the binding affinity of MVSS with human immunoreceptors, the Cluspro 2.0 server (http://cluspro.bu.edu/login.php) (21 October 2022) [47] was applied for molecular docking analysis such as BCR (PDB ID; 5drw), MHC I (PBD ID; 4u6y), MHC II (PBD ID; 5jlz), TLR2 (PDB ID; 2z7x), TLR3 (PDB ID; 3ulv) and TLR4 (PDB ID; 4g8a). PyMOL was employed to visualize the docking complexes. Further, PDBsum was used to map the interacting residues between MVSS and immunoreceptors [48]. In addition, the iMODS server (https://imods.iqfr.csic.es/) (22 October 2022) was used to perform molecular dynamics analysis to assess the binding stability of MVSS with human immunoreceptors [49].

### 2.8. Immune Response Simulation

To evaluate potential immune responses to the vaccine, the C-IMMSIM server (https://kraken.iac.rm.cnr.it/C-IMMSIM/) (22 October 2022) was used to simulate possible immune responses [50]. The three injections in the simulation stage were given time steps of 1, 84, and 168, one of which was set to 8 h. Other settings were defaults.

### 2.9. Expression and Purification of Recombinant Protein MVSS (rMVSS)

The recombinant vector composed of MVSS and pET28a(+) was synthesized by Nanjing GenScript and then transformed into *E. coli* BL21[DE3], cultured in Luria broth (LB) to log phase and then combined with 1 mM Isopropyl-β-D-Thiogalactopyranoside (IPTG), grown at 37 °C 5% CO_2_ for 5 h, and finally, purified by His Ni high-performance chromatography column (GE Healthcare, Chicago, IL, USA). Purified proteins were separated by SDS PAGE on 12.5% gels (Vazyme, Nanjing, China) and stained with Coomassie Blue. Protein concentration was quantified by a BCA kit (TaKaRa, Beijing, China).

### 2.10. Bacterial Strains and Culture

SS2 strain P1/7 was isolated in the UK and is a strongly virulent reference strain of *S. suis* [51]; SS5 strain HN105, SS7 strain 2018WUSS020, SS9 strain GZ0565, SS Chz strain CZ130302, and SS2 strains D74-2 were successively isolated in China [12,14,52,53]. The strains required for the experiment were conserved in our laboratory. All *S. suis* were cultured in Todd Hewitt broth (THB) (Hopebio, Qingdao, China) or THB dish supplemented with 5% (*v*/*v*) sheep’s blood (Maojie, Nanjing, China) at 37 °C with 5% CO_2_. BL21 containing the recombinant vector was cultured in LB medium with Kanamycin at 37 °C.

### 2.11. Mouse Vaccination and Specific Antibody Detection

Four-week-old female ICR mice (n = 72) were randomly and equally divided into two groups and injected subcutaneously with either rMVSS (20 µg/mouse) or an equal volume of PBS mixed with Montanide ISA206 adjuvant (Seppic, Paris, France) three times, 14 days apart. Mice (n = 54) equally selected from both groups were infected with multiple serotypes of *S. suis* to test rMVSS protection rate, and the remaining mice (n = 18) were used for detection of organ bacterial load. Serum was collected from orbital blood of immunized mice (n = 3) before each immunization and 10 d after triple immunization. In addition, to determine vaccine immunity, mice (n = 3) were used for a single immunization, whereby a single orbital blood collection was performed at 15, 29, and 43 d after the first vaccination, and no repeat immunizations were performed. To minimize pain and mortality in mice, the orbital blood collection procedure strictly complied with the guidelines of the National Centre for Replacement, Refinement and Reduction of Animals in Research (NC3Rs) and published articles [54,55]. Briefly, 200 µL of whole blood was collected from the retro-orbital sinuses of anesthetized mice (n = 3) using non-heparinized capillary puncturing and immediately transferred to a 1.5 mL sterile centrifuge tube at 37 °C for 1 h, followed by overnight at 4 °C to separate the serum. After blood collection, the capillary was gently pulled out, the eyelids were closed, and a cotton pad was lightly pressed to minimize bleeding. During blood collection, the mice did not show any signs of consciousness during the sampling procedure. Indirect enzyme-linked immunosorbent assay (ELISA) was used to determine the anti-rMVSS antibody titers. Briefly, rMVSS was measured at a final concentration of 5 µg/mL in sodium carbonate buffer (pH 9.6) with 100 µL encapsulated microliter plates, which were blocked with 200 µL of 0.5% BSA in PBST for 1 h at 37 °C. Serum samples were mixed (1:200) in 0.5% BSA blocking solution, and 100 µL was added to microliter plates and incubated at 37 °C for 1 h. Subsequently, horseradish peroxidase-coupled goat anti-mouse IgG antibody (1:2000) was added and incubated at 37 °C for 1 h. Between each step, the samples were washed with 200 µL PBST for 10 min three times. Antibody binding assay was performed using protein peroxidase coupling (Sigma, P8651, Burlington, MA, USA) and substrate tetramethylbenzidine. The plates were developed with a tetramethylbenzidine (TMB) substrate (Sigma, P8651). Absorbance was measured at 450 nm.

### 2.12. Western Blot Analysis

Western blot (WB) was taken to analyze immune serum reactivity to rMVSS. Briefly, purified rMVSS was separated on 12.5% SDS-PAGE gel and transferred to polyvinylidene difluoride (PVDF) membranes with 5% skimmed milk and held at 37 °C for 1 h. Further, PVDF membranes were incubated with the 1:10,000 dilution serum overnight at 4 °C and then treated with the 1:2000 dilution horseradish peroxidase (HRP)-coupled goat anti-mouse IgG. The membrane was washed with PBS containing 0.05% Tween 20 for 10 min between each step and repeated three times. Finally, the results were observed using Amersham ECL Plus Western blotting detection reagents (GE Healthcare).

### 2.13. Mice Challenge

Mice were infected with SS Chz strain at 1 × 10^7^ cfu/mouse (n = 26) and 5 × 10^7^ cfu/mouse (n = 20) and with SS2 (D74-2) strain at 3 × 10^8^ cfu/mouse (n = 26) after triple immunization. The mice were observed for one week to monitor mortality. 

### 2.14. Bacterial Load Monitoring in Mouse Organs 

Mice (n = 24) were dislocated and executed after CO_2_ sedation after a 12 h observation; blood was collected aseptically and diluted with PBS for inoculation on THB agar plates. In addition, the livers, spleens and brains were collected in MP tubes and homogenized by adding PBS in equal proportions via MP instrument. A portion of the homogenate was diluted with PBS and inoculated in THB agar plates, the rest was used as samples for RNA extraction. All THB agar plates were incubated overnight and colony-forming units (CFU) in the range of 30–300 were counted in the next day; unpaired two-tailed *t*-tests were used to analyze the differences between groups.

### 2.15. Cytokine Assay

Tissue RNA was extracted and reverse transcription quantitative real-time quantitative PCR (RT-qPCR) was performed using FastPure Cell/Tissue Total RNA Isolation Kit V2 (Vazyme Biotech Co., Ltd., Nanjing, China) according to the instructions. RNA was reverse transcribed to cDNA using HiScript II Q RT SuperMix (Vazyme, Nanjing, China). ChamQ SYBR qPCR Master Mix (Vazyme China) combined with QuantStudio 6 Flex instrument (Thermo Fisher Scientific, Shanghai, China) was used to verify the transcript levels of cytokines (IL-1β, IL-2, IL-6, IL-10, and TNFα) in cDNA. The primers used in this study were following published article s [56] and are shown in Appendix A (Appendix A). Reaction parameters were obtained from the qPCR Master Mix kit instructions. Briefly, the first stage is an initial denaturation program of 30 s at 95 °C, the second stage is 40 cycling reactions of 10 s at 95 °C and 30 s at 60 °C, and the final stage is melting curve of 30 s at 95 °C, 30 s at 60 °C, and 15 s at 95 °C. The transcript levels of the housekeeping gene GAPDH were used to normalize the transcript levels of target genes. Relative fold changes were calculated using the 2^−ΔΔCT^ method. At least three replications were performed for each sample. The 2^−ΔΔCT^ method was used to calculate the relative fold change. Each sample was subjected to at least three replications. The primers used in this study are shown in Appendix A (Appendix A).

### 2.16. Histopathological Analysis

The spleens, brains and livers of mice were fixed with 4% paraformaldehyde, embedded in paraffin, cut into 5 µm thin slices, fixed on the glass, stained with hematoxylin/eosin (H&E), and observed under a light microscope.

### 2.17. In Vitro Antimicrobial Assay

In vitro antimicrobial assays were used to determine rMVSS antibody antimicrobial activity against multiple serotypes of *S. suis* strains. The detailed steps are described in our previous study [56]. Briefly, activated *S. suis* was selected and incubated in THB overnight. The strains were transferred to 5 mL THB according to 1:100 and incubated at 180 rpm at 37 °C until log phase (OD600 = 0.6~0.8). Pipettes of 100 μL of various serotypes of *S. suis* were diluted 50-fold with THB solution and transferred to a microtiter plate. rMVSS high immune serum and negative serum were diluted 50-fold with THB; then, 100 μL was added to each well, incubated for 4 h at 37 °C, and then diluted with PBS to inoculate on THB agar plates, which were incubated overnight at 37 °C; the CFUs were counted the next day. The negative serum group was used as a baseline and the experiment was repeated three times.

### 2.18. Passive Immunization

In order to observe the preventive and neutralizing effect of hyperimmune serum in vivo, mice (n = 13) were intraperitoneally injected with 200 μL of anti-rMVSS serum and then infected with 1 × 10^7^ cfu/mouse SSChz CZ130302 after 24 h. Control mice were injected with an equal amount of PBS and SSChz CZ130302. The mortality rate was recorded.

### 2.19. Statistical Analysis

Statistical analysis was performed using GraphPad Prism version 8.0 (GraphPad, La Jolla, CA, USA). Survival curves were analyzed using the Log-rank (Mantel–Cox) test. ELISA was analyzed using one-way ANOVA. Unpaired two-tailed *t*-tests were used for other wet experiments. All tests were considered statistically significant with a *p* value < 0.05 (ns *p* ≥ 0.05, * *p* < 0.05, ** *p* < 0.01, *** *p* < 0.001).

## 3. Result

### 3.1. Pan-Genomic Analysis of the Whole Genome from S. suis

To screen for candidate proteins capable of producing cross-protection, all complete sequenced *S. suis* genomes (n = 120) were downloaded from the NCBI database, all strains were annotated using the native software Prokka, and the output GFF3 files were subsequently pan-genomic analyzed using Roary. The results showed that a total of 17,737 CDSs were identified in the whole genome of *S. suis*, among which the core genome (262) and soft-core genome (615) accounted for approximately 1.48% and 3.47%, respectively (Figure 2A). As more strains increase, the number of conserved genomes decreases and stabilizes, while new genomes could accumulate all the time, suggesting that *S. suis* has an “open pan-genome” (Figure 2B,C). In addition, the diversity of the current completely sequenced *S. suis* genome was revealed by a complete genome phylogenetic tree with a matrix comparison of the presence or absence of core and accessory genes (Figure 2D).

### 3.2. Candidate Protein Screening

To search for highly conserved and immunogenic candidate proteins, we selected 877 genomes from all core and soft-core genomes submitted to the VaxiJen online server. Among the 877 genomes, 87 genomes were considered protective antigens after antigenicity prediction and scored ≥ 95%. Subsequently, promising antigens were uploaded to PSORTb online servers and were used to predict subcellular localization. The results showed that 15 antigens were predicted to be “non-intracellular”. To prove the recombinant vaccine, we combined the findings of other groups and finally screened seven homologous proteins from 15 candidate proteins that were described in immunogenicity papers or are important virulence-associated proteins. To ensure sufficient immunoprotection of the vaccine, we added two classic candidate proteins, MRP and EF. In summary, we finally screened nine proteins for subsequent epitope screening using subtractive proteomics and immunoinformatics (Figure 3A).

### 3.3. Candidate Epitopes Filtering

A combination of Alphafold2 and IEDB predicted T-cell epitopes and B-cell epitopes in nine candidate proteins. Finally, we identified 16 peptides that contained more than two types of epitopes at the same time. To ensure that the candidate epitopes could produce sufficient immunogenicity, VaxiJen was used to test for antigenicity and ultimately selected epitopes with scores > 0.9 for developing a vaccine. In conclusion, we chose 10 epitopes from 6 proteins for the subsequent build up of the multi-epitope vaccine after epitope forecasting for the 9 candidate proteins (Table 1).

### 3.4. Multi-Epitope Vaccine Design

To efficiently expose each epitope of the vaccine and thus induce strong immune responses, two pre-validated linkers, GPGPG and LRMKLPKS, were used to connect the candidate epitopes [56]. In order to maximize the expression and biological activity of the multi-epitope recombinant protein, hybridization of candidate peptides was predicted, and 10 epitopes were linked by GPGPG and LRMKLPKS according to the principle of increasing hydrophilicity from the middle to both sides. The protein sequence of MVSS is displayed in Appendix A (Appendix A). MVSS was predicted to have an antigenicity of 1.1909, and the online server predicted no allergenicity, which could be used for subsequent vaccine development. The protein sequence of MVSS was constructed using an I-TASSER with five 3D models, among which the 3D model with the highest C-score (−2.43) was selected and uploaded to the GalaxyWeb server for further refinement. The refined MVSS 3D structure presented a cylinder structure to enable full epitope exposure (Figure 3B), and the scoring of the Ramachandran plot shows 73.2% residues in most favored regions, 20% residues in additional allowed regions, 3% residues in generously allowed regions, and 3.8% residues in disallowed regions (Appendix A).

### 3.5. Molecular Docking

To characterize the affinity of MVSS for human immune receptors, ClusPro 2.0 was used for molecular docking. Figure 4A,D represent binding models of MVSS with BCR and TLR2, respectively. Figure 4B details hydrogen bond interactions between MVSS and BCR. In addition, hydrogen bond interactions between MVSS and TLR2 were demonstrated in Figure 4E,F. Beyond hydrogen bonds, the PDBsum server was employed to comprehensively understand vaccine–immunoreceptor interactions. The results showed that partial amino acids of the immunoreceptors interacted with MVSS, such as chains A and B of BCR (Figure 4C) and chains A and B of TLR2 (Figure 4G,H), where the interactions included salt bridges, nonbonding contacts, and hydrogen bonds but not disulfide bonds. In addition, MVSS also interacted substantially with chains C and P of MHC-I, chains A-D of MHC-II, chain A of TLR3, and chains B and D of TLR4 (Appendix A). Altogether, these results proved that MVSS has a good affinity for host immune receptors.

### 3.6. Molecular Dynamics Simulation

To assess the molecular dynamics trends of the rMVSS-TLR2 complex in the host, MD simulations of the complex were carried out using the online iMODS server. Deformability was established as the independent deformation of each residue, which was depicted using the chain-hinge method (Appendix A). Appendix A (Appendix A) showed the deformability map of the complex, where the peaks indicated non-rigid regions of the protein. The complex had an eigenvalue of 1.020381 × 10^−5^ (Appendix A). In addition, the covariance matrix plots of the residuals were negatively correlated with the eigenvalues, showing the individual (red) and cumulative (green) variances (Appendix A). The covariance matrix indicated the coupling between residue pairs and represented the correlation experience: red showed correlation, white showed uncorrelation, and blue showed anticorrelation motion (Appendix A). The elastic network of the rMVSS-TLR2 complex is shown in Appendix A (Appendix A), where dots indicate springs and gray regions represent stiffer springs. Overall, the results obtained indicated that rMVSS exhibits modest fluctuations, a compact structure, and sustained binding interactions with TLR2.

### 3.7. In Silico Simulation of Multi-Epitope Vaccine Immunization

To confirm that rMVSS could induce an immune response, in silico immune simulation was performed, and the results show that the antibody titer significantly increased with a second booster vaccination in Appendix A (Appendix A). Appendix A also revealed that immune response-associated cell numbers of B cells, T cells and various cytokines all markedly improved after immunization. Among them, IFN-γ, an indicator of the Th1-type immune response, is an important assessment for the development of porcine streptococcal vaccines [23,57], and it was strongly induced in rMVSS-simulated immunization. In addition IL-2, which is associated with the prevention of pathogenic bacterial infections [58], was also strongly stimulated to be upregulated with multiple immunizations against MVSS. In conclusion, these results illustrated that MVSS could effectively elicit a strong immune response.

### 3.8. Expression and Purification of rMVSS

MVSS with 330 amino acids was artificially synthesized after codon preference optimization and was ligated with pET-28a(+) to construct the recombinant plasmid pET-28a-MVSS (Appendix A). The recombinant plasmid pET-28a-MVSS was introduced into the expression bacterium BL21 (Figure 5A), and rMVSS was purified using a HisTrap nickel column after induced expression. SDS-PAGE electrophoresis showed that rMVSS had 45 kDa in the supernatant, which changed to a single band under the effect of an eluent containing 500 mM of imidazole (Figure 5B).

### 3.9. Immunogenicity of rMVSS in Mice

The immunization procedures of rMVSS-immunized mice are shown in Figure 5C. WB displayed specific bands similar to rMVSS in SDS-PAGE electrophoresis around 45 kDa, demonstrating that the mice serum, after triple immunization, produced an anti-rMVSS polyclonal antibody (Figure 5D). In addition, unprocessed-produced images of all the bands on the Western blot are shown in Appendix A. Meanwhile, the indirect ELISA results showed a significant increase in anti-rMVSS polyclonal antibody titers in the mouse serum after triple immunization compared with the negative serum (Figure 5E). Interestingly, although anti-rMVSS polyclonal antibodies were equally elevated with the number of immunizations (Figure 5E), the serum antibody titers after one immunization were significantly increased compared with the negative serum (Figure 5F).

### 3.10. Mice Challenge Studies

To determine rMVSS immunoprotection against *S. suis* infection, we challenged mice with *S. suis* after triple immunization. Owing to the high pathogenicity of *S. suis* CZ130302 in mice models [14,15], we first injected mice with strain CZ130302 1 × 10^7^cfu/mouse after three immunizations and observed for seven days. The results showed a survival rate of 40% (4/10 mice) in the sham-immunized group and 70% (7/10 mice) in the rMVSS-immunized group with a 50% protection rate (Figure 6A). Although no statistically significant difference was found between the two groups of mice, the bacterial loads in the blood, brains, livers and spleens of rMVSS-immunized mice (n = 3) were significantly decreased compared with sham-immunized mice (n = 3) (Figure 6C), demonstrating that the rMVSS multi-epitope vaccine played a role in impeding susceptibility to infection by *S. suis*. Spleen cytokine levels were monitored, and the rMVSS-immunized mice produced higher levels of cytokines IL-1β, IL-6, IL-10 and TNFα compared to the sham-immunized mice (Figure 6E).

To better estimate the MVSS vaccine’s effect on *S. suis* infections, a high dose of strain CZ130302 was injected into mice after triple immunization (5 × 10^7^ cfu/mouse). The results showed that the survival rate in the sham-immunized group was 0 (0/7 mice), and all mice died within 90 h. The survival rate in the rMVSS-immunized group was 14% (1/7 mice) (Figure 6B). Although there was no significant difference between the two groups, mice had a delayed time to death in the rMVSS-immunized group, suggesting that rMVSA was effective in delaying the invasion of *S. suis* and was immune-protective after a high-dose attack (Figure 6B). Compared with the sham-immunized mice (n = 3), the bacterial loads in the blood, brains, livers and spleens were significantly decreased in the rMVSS-immunized mice (n = 3) (Figure 6D). In contrast to the sham-immunized mice’s spleens, lymphocytes in the rMVSS-immunized mice produced higher levels of cytokines IL-1β, IL-6, IL-10 and TNFα (Figure 6F). In conclusion, high- and low-dose *S. suis*-infected mouse assays demonstrated that MVSS induced high titer protective antibodies against *S. suis* infection.

To evaluate the cross-protectiveness of the MVSS vaccine against *S. suis* infections, triple immunized mice were similarly infected with SS2 strain D74-2 (3 × 10^8^ cfu/mouse). The results showed that the survival rate was 50% (5/10 mice) in the sham-immunized group and 70% (7/10 mice) in the rMVSS-immunized group, with a 40% protection rate (Figure 7A). Importantly, the surviving rMVSS-immunized mice had no significant sequelae, whereas all surviving sham-immunized mice showed apparent binocular blindness (Figure 7B). Similarly, the bacterial loads in blood, brains, livers and spleens were decreased in the rMVSS-immunized mice (n = 3) (Figure 7C). Compared to sham-immunized mice, the rMVSS-immunized group produced higher levels of cytokines IL-1β, IL-2, IL-6, IL-10 and TNFα in the spleen (Figure 7D). In addition, histopathology was used to observe the lesions in all infected mice organs. Pathological sections showed that compared with rMVSS-immunized mice, all sham-immunized mice had severe hemorrhages in the brains, spleens, and livers (Appendix A). In conclusion, the mice models proved that the MVSS vaccine acted as a barrier to SS2 infections and improved healing in infected mice.

### 3.11. Antimicrobial Activity of Anti-rMVSS Polyclonal Antibody

Considering the serotype diversity of *S. suis* and the pathogenicity variability in mice models, the antibody inhibition assay is regarded as a simple and effective method for evaluating in vitro antimicrobial activity of the rMVSS antibody against multi-serotype *S. suis*. rMVSS polyclonal serum was mixed with five potentially infectious human serotypes of *S. suis* strains (SS2, SS5, SS7, SS9, and SSChz), and bacterial replicates was detected by viable colony counting after incubation for 4 h. The results suggested that the rMVSS immune serum significantly inhibited the growth of different serotypes of *S. suis* strains (Figure 8A).

## 4. Passive Immunity

To further assess the MVSS potential, anti-rMVSS serums were tested for passive immunity against *S. suis*. Mice were infected with SSChz (1 × 10^7^ cfu/mouse) after anti-rMVSS serum treatment. The results showed that the survival rate of untreated mice was 60% (6/10 mice), and the survival rate of anti-rMVSS-treated mice was 100% (10/10 mice) (Figure 8B). Additionally, anti-rMVSS-treated mice (n = 3) had fewer bacteria loads in the blood, brains, livers, and spleens than PBS-treated mice (n = 3) after *S. suis* infection (Figure 8C). In addition, the RT-qPCR assay indicated that anti-rMVSS-treated mice had significantly higher spleen cytokine IL-1β, IL-2, IL-6, IL-10 and TNFα transcript levels than the PBS-treated mice (Figure 8D). In comparison with PBS-treated mice, anti-rMVSS-treated mice had intact meninges, with no apparent hemorrhages in brains and livers (Figure 8E). In conclusion, these results confirmed that the anti-rMVSS serum was able to provide positive immunoprotection to hosts infected with *S. suis*.

## 5. Discussion

The increasing drug resistance of *S. suis* has forced vaccines to become the most cost-effective measure to prevent human *S. suis* infections [18]. Multi-epitope vaccines are considered a desirable vaccine type owing to their broad coverage and robust immunoprotection. In this study, the multi-epitope vaccine MVSS was designed based on ten candidate epitopes selected from six highly conserved proteins. Both rMVSS-immunized and passive immunization using an anti-rMVSS serum prevented a lethal dose of *S. suis* infection, induced high antibody titers, stimulated cytokine expression, and attenuated pathological damage caused by *S. suis* in mice. In addition, in vitro assays confirmed that the anti-rMVSS serum exerted growth inhibitory effects against various important serotypes of *S. suis*, reflecting the cross-protective potential of MVSS against *S. suis* infections and promising to be an effective vaccine candidate for millions of people with occupational contact with pigs and pork against *S. suis* infections.

Compared with traditional inactivated vaccines and emerging mRNA vaccines, multi-epitope vaccines present enhanced immune efficacy and broad-spectrum protection and are low cost [59,60]. Candidate epitope screening determines immunological efficacy and protective coverage, and therefore, highly conserved and immunogenic epitopes need to be prioritized [24,27]. With the development of immunoinformatics, in silico simulations can greatly facilitate vaccine design [61,62]. For example, COVID-19 vaccines and human papillomavirus vaccines developed via bioinformatics techniques performed well in in silico immune response simulations [63,64]. In addition, the commercially available vaccine [Bexsero^®^] [65], which has been used in humans to prevent *Neisseria meningitidis* serogroup B infections, and a multi-epitope vaccine offering cross-protection against multiple serotypes of *Streptococcus agalatiae*, confirming the viability of a vaccine design based on immunoinformatics [56]. In this study, we also screened six candidate proteins for vaccine design based on immunoinformatics and reverse vaccinology. Importantly, B-cell conformational epitopes were likewise used as filtering requirements in the subsequent epitope screening. Benefiting from the three-dimensional structural information provided by accurate prediction tools, in silico prediction and modeling tools were substantially and significantly improved [66]. Herein, we used Alphafold2 for the 3D modeling of six candidate proteins, which was further combined with the IEDB database considering thresholds greater than −3.7 as valid conformational epitopes. This approach of structural information-predicted conformational epitopes is regarded as fundamental in guiding vaccine development [67].

To guarantee a multi-epitope vaccine with immunoprotective efficacy, MRP and EF, which have been recognized as candidate proteins, were used in vaccine design. MRP and EF are classic virulence factors in *S. suis* and are the main antigens recognized when recovering serum from infected hosts [21]. Compared to vaccines prepared with individual antigens, subunit vaccines combining MRP and EF provide better protection against SS2 attacks [19]. However, their use is limited due to a lack of conservation [15,68]. To ensure the cross-protectiveness of the multi-epitope vaccine, pan-genomics and bioinformatic analysis were used to filter additional subsequent proteins. Enolase (WP_002935704.1), zinc-binding protein AdcA (WP_004195559.1), cell wall protein (WP_ 033875493.1) and penicillin-binding protein 2B (WP_004298861.1) were selected for vaccine design, with marked advantages in terms of conservation, immunogenicity, and subcellular location. Surprisingly, our results based on in silico screening have similarities with experimental findings from other groups. Enolase is responsible for *S. suis* adhesion and invasion into the host, and immunoproteomics and animal experiments have confirmed that enolase is a viable candidate protein [20]. Similarly, the zinc-binding protein AdcA (WP_004195559.1) [69] and cell wall protein (WP_033875493.1) [70] are considered to be immunogenic. Although we currently find no proof that the penicillin-binding protein 2B (WP_004298861.1) has better immunogenicity, several groups have confirmed its location on the *S. suis* surface, which is a virulence-associated factor responsible for *S. suis* growth [71]. In conclusion, we combined online bioinformatic analysis and published experimental data to select six highly conserved proteins with immunogenicity for multi-epitope vaccine MVSS design.

To evaluate MVSS as a candidate vaccine for preventing *S. suis* infections, mouse models were immunized. Interestingly, rMVSS-immunized mice had remarkably higher cytokine levels than sham-immunized mice in splenocytes. This may be related to the “GPGPG” and “LRMKLPKS” linkers in rMVSS. “GPGPG” and “LRMKLPKS” induced intense immune responses by ensuring correct epitope exposure and enhancing MHCII presentation, respectively [72,73]. Similarly, the multi-epitope *Streptococcus pneumonia* vaccine and multi-epitope *Streptococcus agalatiae* vaccine using the same linker elicited high immune responses [56]. In this study, two linkers helped MVSS to expose each epitope efficiently, which may have allowed high-titer protective antibody sera to be produced in mice despite only one immunization. Further, immunized mice were first infected with two important serotypes, SS2 and SSChz. Admittedly, rMVSS-immunized mice showed delayed mortality and better healing than sham-immunized mice but failed to fully survive. In addition, certain indicators that provide vaccine efficacy may be phenotypically different in mouse models than in humans for in vivo tissue responses. Nonetheless, anti-rMVSS serum inhibited five potentially human-infectious *S. suis* serotypes in vitro, confirming that MVSS was a hopeful vaccine candidate for preventing human Streptococcus suis infections. Moreover, in the immunity protection against *S. suis* infection models, the mice model has been shown to excellently reproduce the typical clinical signs of *S. suis* diseases, including septicemia, meningitis and infectious shock, thus making it a cost-effective, convenient and commonly used model [74,75]. Therefore, optimizing vaccine immunization, adjuvant selection, and immunization procedures will be considered in subsequent experiments.

## 6. Conclusions

In conclusion, our study partially filled the gap regarding multi-epitope vaccines for the prevention and treatment of *S. suis* infections and provided new insight for combating human infections with *S. suis* diseases in the future. In addition, the availability of a multi-epitope vaccine design pipeline through pan-genomics and bioinformatics inspired multi-valent vaccines designed against other multi-serotype bacterial pathogenic diseases.

## Figures and Tables

**Figure 1 vaccines-12-00137-f001:**
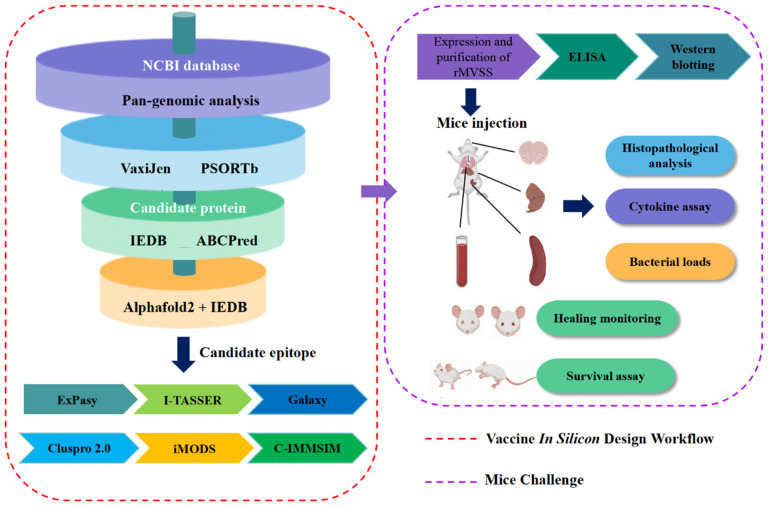
The workflow that was employed in silico immunoinformatics analysis combined with mice challenge to design a multi-epitope vaccine against *S. suis*.

**Figure 2 vaccines-12-00137-f002:**
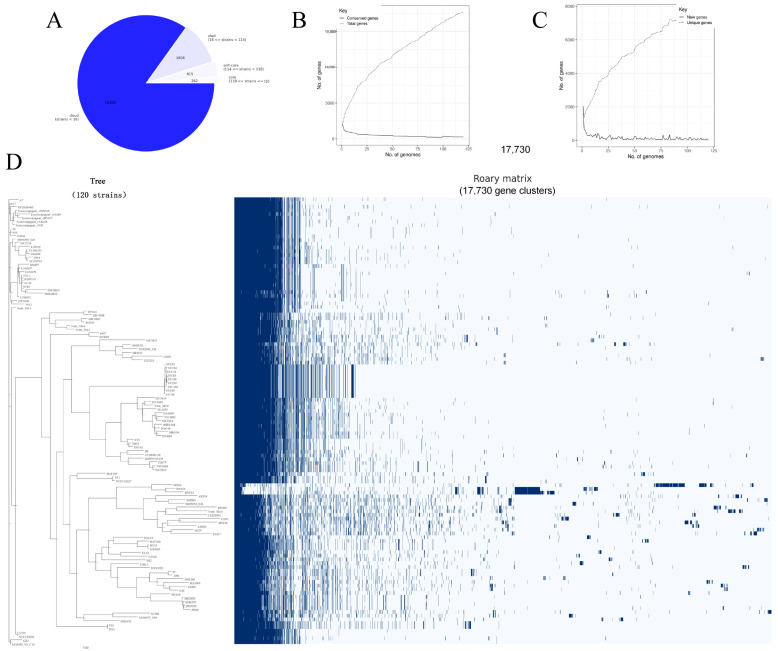
Pan-genomic analysis of 120 *S. suis* strains. (**A**) Pan-genomic analysis identified the distribution of genes in *S. suis* genome. (**B**) Total genes significantly increased as the amount of *S. suis* grew, whereas conserved genes gradually dropped and stabilized. (**C**) Quantities of new and unique genes in *S. suis* were positively and inversely proportional to the number of isolate strains, respectively. (**D**) Whole-genome phylogenetic tree and matrix of gene presence and absence. On the left is the phylogenetic tree constructed based on core genome comparison to show the evolutionary relationships among all *S. suis* genomes; on the right is the matrix constructed based on the presence or absence of genes to display the clustering of genes in all *S. suis* genomes. Core and soft-core genes in red brackets were used in the candidate protein screening.

**Figure 3 vaccines-12-00137-f003:**
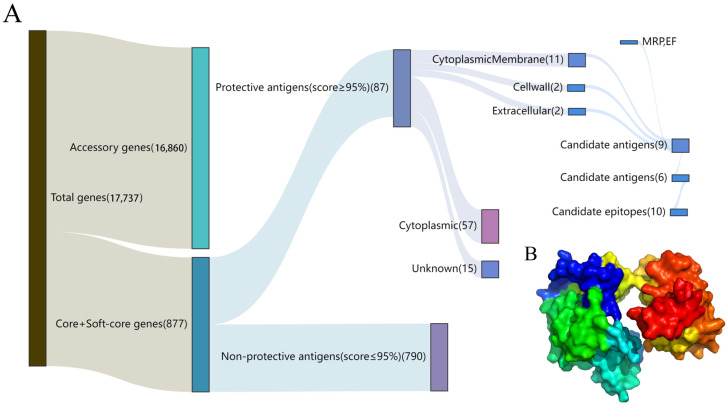
Subtractive proteomics of the core proteome. (**A**) Ten candidate epitopes were filtered for vaccine design by protective antigen prediction, subcellular localization and B/T cell epitope screening. (**B**) MVSS 3D modelling. Different epitopes were indicated by various colors.

**Figure 4 vaccines-12-00137-f004:**
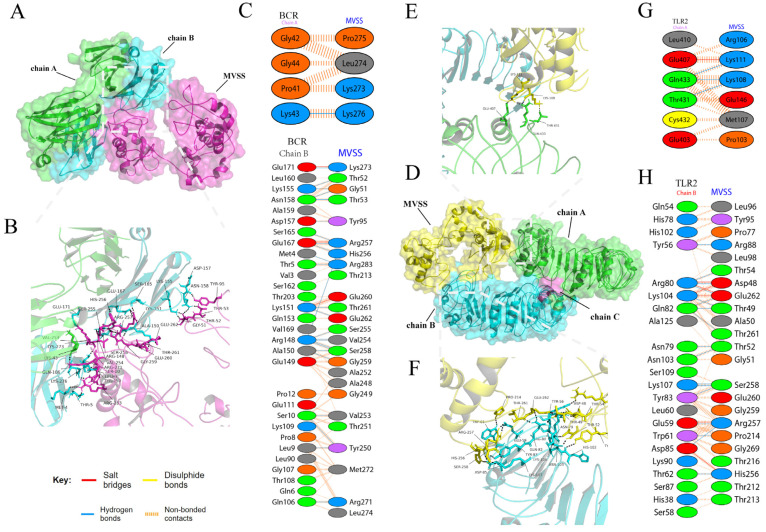
Schematic diagram of molecular docking for MVSS and human immune receptors. (**A**) Illustration of the docking of the MVSS-BCR complex. (**B**) Graphic representation of the docking conformation and hydrogen bonding interactions between MVSS (purple) and BCR chains A (green) and B (blue); black dashed lines indicate hydrogen bonds. (**C**) Residues of MVSS interacting with BCR. (**D**) Docking schematic of the MVSS-TLR2 complex. Schematic diagram of the docking conformation and hydrogen bonding interactions of MVSS (yellow) and BCR chains A (green) (**E**) and B (blue) (**F**); black dashed lines indicate hydrogen bonding. Residues of MVSS interacting with BCR chains A (**G**) and B (**H**).

**Figure 5 vaccines-12-00137-f005:**
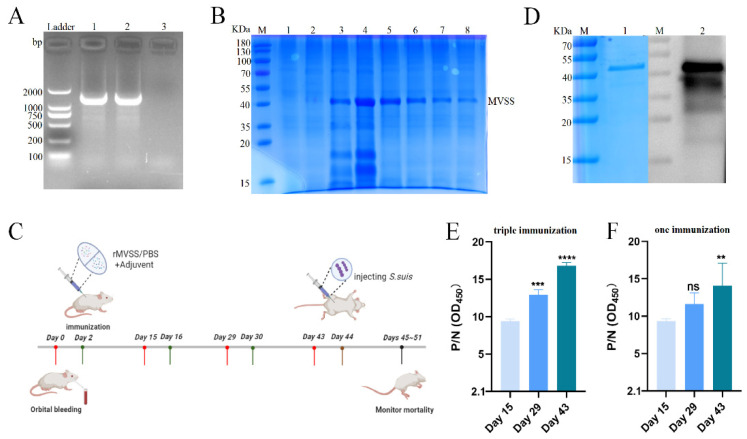
Recombinant protein MVSS expression, purification and immunization (**A**) Ladder as indicator, lane 1: pET-28a-MVSS-BL21; lane 2: pET-28a-MVSA; lane 3: negative. (**B**) M is Marker, lanes 1–2 are pre-induction, lanes 3–4 are post-induction, and lanes 5–8 are post-purification. (**C**) MVSS immunization of mice procedure. Mice were divided into rMVSS-immunized and sham-immunized control groups. Red columns are blood collected, green columns are immunized, brown columns are injected with *S. suis*, and black columns are monitored mortality. Schematic material sourced from BioRender. (**D**) M indicates Marker, lane 1 refers to SDS-PAGE gel purified with Ni-NTA agarose for rMVSS protein, and lane 2 is western blot to detect the reactivity of anti-rMVSS serum with rMVSS. (**E**) Antibody titer induced by rMVSS after triple immunization. (**F**) Antibody titer induced by rMVSS at 15 d, 29 d and 43 d after only one immunization. ELISA was analyzed using one-way ANOVA (ns *p* ≥ 0.05, ** *p* < 0.01, *** *p* < 0.001, **** *p* < 0.0001).

**Figure 6 vaccines-12-00137-f006:**
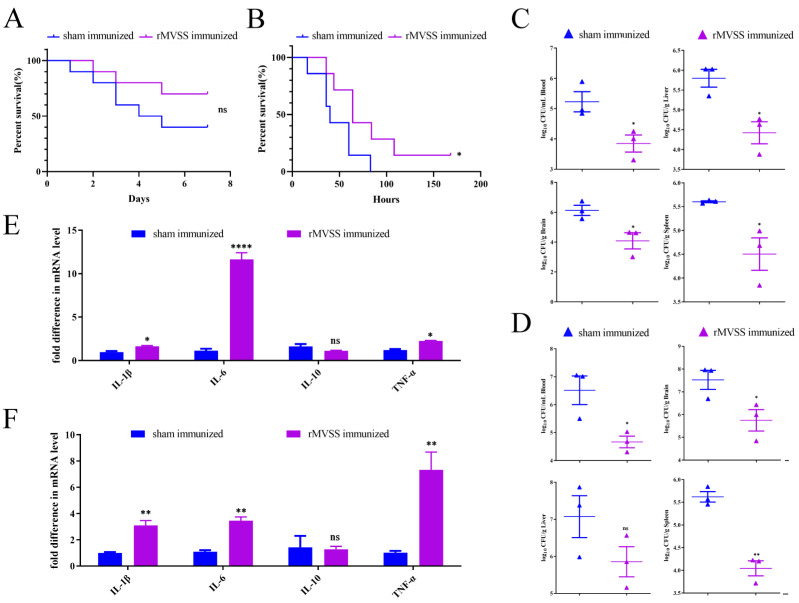
Evaluation of MVSS active immunization protection of mice infected with SSChz strain. Survival curves of rMVSS-immunized and sham-immunized mice infected with SSChz, 1 × 10^7^ cfu/mouse (**A**) (ns, *p* = 0.1751) (sham immunized: 4/10 mice; rMVSS immunized: 7/10 mice) and 5 × 10^7^ cfu/mouse (**B**) (*, *p* = 0.0374) (sham immunized: 0/7 mice; rMVSS immunized: 1/7 mice). Bacterial counts in mice organs in different groups after infection with SSChz, (**C**) low dose (**D**) high dose. (**F**) Cytokine levels in splenocytes of mice after infection with strain CZ130302, (**E**) low dose (**F**) high dose. Survival curves were analyzed by log-rank (Mantel–Cox) test and the rest by unpaired two-tailed Student’s *t*-test (*, *p* < 0.05; **, *p* < 0.01; ****, *p* < 0.0001; ns, not significant).

**Figure 7 vaccines-12-00137-f007:**
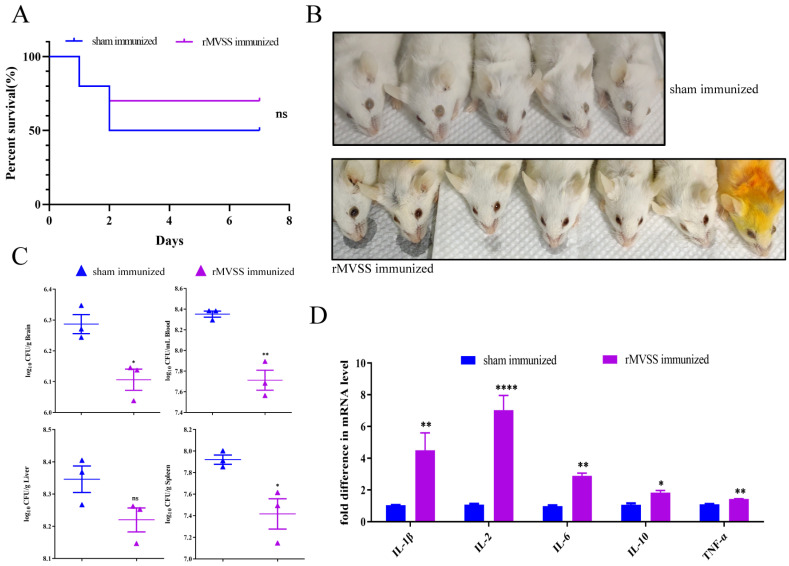
Assessment of MVSS active immunization protection of mice infected with SS2 strain. (**A**) Survival curves of SS2 D74-2 strain infected rMVSS-immunized (7/10 mice) and sham-immunized mice (5/10 mice) (3 × 10^8^ cfu/mouse) (ns, *p* = 0.4352). (**B**) Symptoms of SS2 D74-2 infected rMVSS-immunized (n = 7) and sham-immunized mice (n = 5). The upper image shows abnormal eye color, showing white in sham-immunized mice group, and the lower image shows normal eye color in rMVSS-immunized mice group. (**C**) Bacterial counts of mice organs in different groups after infection with SS2 D74-2. (**D**) Cytokine levels in splenocytes of mice in both groups after infestation with SS2 D74-2. The log-rank (Mantel–Cox) was used to examine survival curves, and the unpaired two-tailed Student’s *t*-test was used to assess the remaining data (*, *p* < 0.05; **, *p* < 0.01; ****, *p* < 0.0001; ns, not significant).

**Figure 8 vaccines-12-00137-f008:**
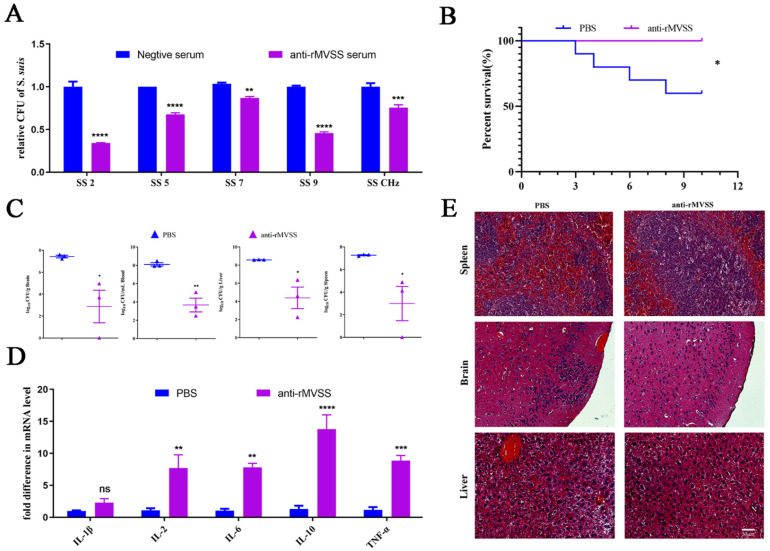
MVSS evaluation of passive immunity. (**A**) Antibacterial activity of anti-rMVSS serum against multiple serotyped *S. suis* strains. (**B**) Survival curves of anti-rMVSS serum immunized (10/10 mice) and PBS-immunized mice (4/10 mice) infected with SSChz (1 × 10^7^ cfu/mouse) (*, *p* = 0.0291). Organ bacterial counts (**C**), cytokine levels in splenocytes (**D**), and histopathological analysis (**E**) in anti-rMVSS serum-immunized and PBS-immunized mice infected with SSChz. Black lines indicate bleeding areas. Survival curves were examined using the log-rank (Mantel–Cox) test, while the remaining data were subjected to an unpaired, two-tailed Student’s *t*-test (*, *p* < 0.05; **, *p* < 0.01; ***, *p* < 0.001; ****, *p* < 0.0001; ns, not significant).

**Table 1 vaccines-12-00137-t001:** Ten epitopes were chosen using immunoinformatics from six candidate proteins.

GenBank	Annotation	Selected Epitope	Antigenicity Score	Hydropathicity
ACS66679.1	Enolase	AKEAGYTAVVSHRSGETEDS	1.2969	−0.885
		GEHEAVELRDGDKSRYLGLG	0.9193	−0.995
WP_033875493.1	Cell wall protein	GDTAGTTTDTKTPEKANDGG	2.273	−1.455
		EKGVNAIVVLAHVPATSKDG	0.9026	0.255
WP_004298861.1	Penicillin-binding protein 2B	LNILFSIVIFLFLVLILRLA	2.0369	2.72
		NGPRTEINMKKRKNKPLEHD	0.9815	−2.145
WP_004195559.1	hypothetical protein	GEEEHEGHDHSEEGHSHAYD	1.7881	−2.315
CAA50714.1	extracellular protein factor (EF)	IAGYRTVNSDGTKTETVEET	1.5104	−0.88
ACN96609.1	muramidase-released protein (MRP)	TTPGTNGEVPNIPYVPGYTP	1.1127	−0.61
		TKDGLRYVLVPSKTDGEENG	1.0219	−1.005

## Data Availability

The whole genome sequences used in this study were available from the NCBI database. All data generated or analyzed in this study have been submitted with this paper. Therefore, all data from this study are publicly available.

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
