# Peer review of "Combined Immunoinformatics to Design and Evaluate a Multi-Epitope Vaccine Candidate against Streptococcus suis Infection"

_vaccines, 2024, doi:10.3390/vaccines12020137_

Round 1

Reviewer 1 Report

Comments and Suggestions for Authors

Nice topic and work and thanks for the focus on this very important topic and area, but it needs substantial improvement; some issues related to the methodology, repeatability and reproducibility specially the in vivo part, the quality of figs etc…. should be improved. The novelty of the work should be better addressed appropriately in the revised version.

-The conclusion part of the abstract is weak and introductory style. Please elaborate in a very strong manner.

-Introduction should be improved and some points related the use/application and novelty of such vaccine in swine?

-At the end of your introduction the point “In conclusion, our study partially filled the gap of multi-epitope vaccines for the prevention and treatment of S. suis infections and provided new insight for combating human infections with S. suis diseases in the future”, should be removed from here and better used in the finding and the conclusion part of the manuscript.

-Some words throughout the text should be appeared correctly ... like “cytoplasmicmembrane” in line 109 or line 140...”C-score scores were”… etc... please carefully recheck throughout the entire text.

-Methods should be hugely improved and appropriately elaborated.

- Throughout the text of the methods, it is unnecessary to rewrite the notion “Four-week-old female ICR mice”... once is okay, like your observational fig. The experimental design should be highlighted in the written text as well; you should first explain the experimental design and here you can mention the numeration etc….

-For your design on TLR2 and TLR4 on the in silico part, please briefly address which part(s) of these molecules were targeted for the in silico based analyses?; the same for MHC-II?

-About the colony-forming units (CFU), please clearly mention how it was counted and analyzed?

-In the methods, please clearly mention the name of the cytokines in your transcriptional analyses? And also the detailed procedure how it was optimized, specially annealing temperature and the time of the steps of the cycle of the qPCR protocol; also, the primers were newly designed or from other papers? Please elaborate very clearly in the text. Readers might want to see and use.

-You mean the rMVSS antibody possess antibacterial activity? How comes and what is mechanism? Also the procedure of antimicrobial activity of anti-rMVSS polyclonal antibody after 4 h incubation you counted the CFU? Why only 4 h? is it enough for the assay? At least overnight or more needed for colony formation…?  

-Some of your figures’ quality is poor and hardly readable (specially fig 2, fig 4, also oethers). Please revise and prepare with much higher resolution.

Lines 274- 286, mostly are methods… it should be appeared in the methods… may be a brief and fig. might be appeared in the result section. Please carefully elaborate and appropriately remove form here.

Lines 365-3666, this is not result. Please move to the introduction part.

-It is unclear how much blood was taken? How about the approach of taking blood? Please elaborate in the text. In fig 5D it is written orbital bleeding? Why not other appropriate approach of taking blood samples for suitably acceptable animal welfare committee. I would not use this approach!

-About Fig 5, its caption, your should clearly mention the expression of what? Also marker (M) normally used for proteins… but for 5A you should define and replace “M” by “ladder”. Please elaborate here and throughout the text.

Your device-produced images should be more appropriately appear in the manuscript.

-About the numeration of mice in your in vivo study, the “n” is unclear … for survival rate in relation to “Evaluation of MVSS active immunization protection on mice infected with SSChz strain and others” number of mice? etc…? Results related to the figs 6 and 7 and others? It should be clearly and appropriately mentioned in the text and captions. Please also add much better quality for fig 8E (pathological ones).

Comments on the Quality of English Language

Minor correction

Author Response

Dear Editor and Referees,

Thank you for giving us the opportunity to revise our manuscript.

We have revised and improved the manuscript according to the Referees′ helpful comments. Our responses to each question from the referees are attached below. We appreciate the comments and advice provided by the Editors/Reviewers and hope that the corrections will meet with your approval.
We look forward to reading your appraisal of the revised paper.

Yours sincerely,
Guangjin Liu

Replies to Referee 1

Comments:

Nice topic and work and thanks for the focus on this very important topic and area, but it needs substantial improvement; some issues related to the methodology, repeatability and reproducibility specially the in vivo part, the quality of figs etc…. should be improved. The novelty of the work should be better addressed appropriately in the revised version.

  1. The conclusion part of the abstract is weak and introductory style. Please elaborate in a very strong manner.

The authors’ answer:

Thanks for your comment. In the revised manuscript, these sentences were edited as:

“Given the above, it is potential to develop MVSS as a universal subunit vaccine against multiple serotypes of S. suis infections.”

  1. Introduction should be improved and some points related the use/application and novelty of such vaccine in swine?

The authors’ answer:

Thanks for your comment. In the revised manuscript, we improved the application and novelty of this vaccine in swine in the “Introduction” section, as follow as:

“In addition, the protein JointS, a combination of MRP, glyceraldehyde 3-phosphate dehydrogenase (GAPDH), and the novel dihydrolipoamide dehydrogenase DLD, provided favorable protection in a virulent SS2-infected piglet model[1], which presents an attractive approach: compared to single antigen vaccines, co-expression of multiple antigens provides better protection in livestock vaccines.”

“Recently, a universal multi-epitope candidate vaccine was constructed against swine-infected S. suis using an immunoinformatics approach[2]. However, the limited surface proteins of S. suis and the absence of experimental verifications appear to be insufficient to convincingly validate the availability of the vaccine. Therefore, the current protection data for multi-epitope vaccines against S. suis needs to be expanded.”

  1. At the end of your introduction the point “In conclusion, our study partially filled the gap of multi-epitope vaccines for the prevention and treatment of S. suis infections and provided new insight for combating human infections with S. suis diseases in the future”, should be removed from here and better used in the finding and the conclusion part of the manuscript.

The authors’ answer:

Thank you for your comment. In the revised version, we moved the sentence to the "Conclusion" section for readability.

  1. Some words throughout the text should be appeared correctly ... like “cytoplasmicmembrane” in line 109 or line 140...”C-score scores were”….. please carefully recheck throughout the entire text.

The authors’ answer:

Thanks for your careful checks. In the revised manuscript, we scrutinized the entire text and corrected "cytoplasmicmembrane" to "cytoplasmic membrane", "C-score scores were..." to "C-scores were...", "cellwall" to "cell wall", "wet-experiments" to "wet experiments" and "Maeker" to "Ladder" and so on.

  1. Methods should be hugely improved and appropriately elaborated.

The authors’ answer:

Thanks for your comment. In the revised manuscript, we hugely improved the “Materials and methods” section based on all your comments such as: added description of the number of mice, detailed steps for blood collection and in vitro antimicrobial assay, counting and analysis of CFUs, primers and systems involved in RT-qPCR, and so on. These modifications are also described in the following answer, thanks again for all your comments.

Beyond that, to improve readability, we detailed the in silico analytical tools and experimental protocols (lines 97-265 in the revised manuscript) and summarized as follows: 2.1~2.6 mainly focused on the design of MVSS which was a candidate vaccine against multiple serotypes of S. suis infections, through the screening of conserved proteins and epitopes, and 2.7~2.8 verified its immune efficacy via in silico molecular docking and simulated immunization. Furthermore, 2.9~2.17 were in vitro and in vivo experiments to evaluate the immune potency of MVSS.

  1. Throughout the text of the methods, it is unnecessary to rewrite the notion “Four-week-old female ICR mice”... once is okay, like your observational fig. The experimental design should be highlighted in the written text as well; you shouldfirst explain the experimental design and here you can mention the numeration etc….

The authors’ answer:

Thanks for your suggestions. In the revised version of “Material and Methods”, we retained the description "Four-week-old female ICR mice" only once, as well as explaining the experimental design and adding a description of the number of mice in the text, such as:

“Four-week-old female ICR mice (n=72) were randomly and equally divided into two groups and injected subcutaneously with either rMVSS (20 µg/mouse) or an equal volume of PBS mixed with Montanide ISA206 adjuvant (Seppic, France) three times, 14 days apart. Mice (n=54) equally selected from both groups were infected with multiple serotypes S. suis for testing rMVSS protection rate and the remaining mice (n=18) were used for detection of organ bacterial load.”

  1. For your design on TLR2 and TLR4 on the in silico part, please briefly address which part(s) of these molecules were targeted for the in silico based analyses?; the same for MHC-II?

The authors’ answer:

Thank you for your comments. In the in silico part, the whole amino acids of the immunoreceptors (TLR2, TLR4, and MHC-II, etc.) were used to analyze the affinity with MVSS, and the subsequent molecular docking results showed that the partial amino acids have interactions with the MVSS, such as the chain A and B of TLR2 (Fig. 4), the chain B and D of TLR4 (Fig. S3), chain A-D of MHC-II (Fig. S3). In the revised version, we added a detailed description as follows;

“The results showed that partial amino acids of the immunoreceptors interacted with MVSS, such as chains A and B of BCR (Fig. 4C), and chains A and B of TLR2 (Fig. 4G, H), which interactions included salt bridges, nonbonding contacts, and hydrogen bonds, but not disulfide bonds. In addition, MVSS also interacted substantially with chains C and P of MHC-I, chains A-D of MHC-II, chain A of TLR3, and chains B and D of TLR4 (Figure S3).”

  1. About thecolony-forming units (CFU), please clearly mention how it was counted and analyzed?

The authors’ answer:

Thanks for your suggestion. In the “Material and Methods” section of the revised manuscript, we detailed mentioned for how the colony-forming units (CFU) were counted and analyzed, as follows as, "All THB agar plates were incubated overnight and colony-forming units (CFU) in the range of 30-300 were counted in the next day, unpaired two-tailed t-tests were used to analyze the differences between groups."

  1. In the methods, please clearly mention the name of the cytokines in your transcriptional analyses? And also the detailed procedure how it was optimized, specially annealing temperature and the time of the steps of the cycle of the qPCR protocol; also, the primers were newly designed or from other papers? Please elaborate very clearly in the text. Readers might want to see and use.

The authors’ answer:

Thank you for the friendly reminder. In the revised version, we elaborated the above-mentioned details in the manuscript:

"ChamQ SYBR qPCR Master Mix (Vazyme China) combined with QuantStudio 6 Flex instrument (Thermo Fisher Scientific, China) were used to verify the transcript levels of cytokines (IL-1β, IL-2, IL-6, IL-10, and TNFα) in cDNA. The primers used in this study were referred to published articles[3]and shown in Table S1 (Supplementary materials). Reaction parameters were referred to the qPCR Master Mix kit instructions. Briefly, the first part of 95°C for 30s, followed by 40 cycles of 95°C for 10s, 60°C for 30s, and finally 95°C for 30s, 60°C for 30s, and 95°C for 15s."

  1. You mean the rMVSS antibody possess antibacterial activity? How comes and what is mechanism? Also theprocedure of antimicrobial activity of anti-rMVSS polyclonal antibody after 4 h incubation you counted the CFU? Why only 4 h? is it enough for the assay? At least overnight or more needed for colony formation…?

The authors’ answer:

Thanks for your comments. In our study, rMVSS was composed of multiple epitopes from conserved surface proteins of S. suis, so that the rMVSS antibody could bind to the surface proteins of S. suis. Based on Prakash N. Reddy's study, the binding of Klebsiella pneumoniae surface proteins with antibodies could hamper the survival of Klebsiella pneumoniae[4]. The same phenotype was observed in group B Streptococcus[3]. The detailed procedure of in vitro antimicrobial assays was referenced to the published article[3,4] with minor modifications. We are so sorry that there was an error in the section of "Materials and Methods", S. suis inoculated on THB plates needed to be incubated overnight at 37°C and colony-forming units (CFU) were counted the next day instead of 4h. We have corrected it in the revised manuscript, as follow as:

“rMVSS high immune serum and negative serum were diluted 50-fold with THB and 100 μL was added to each well incubated for 4 h at 37°C and then diluted with PBS to inoculate on THB agar plates, which were incubated overnight at 37°C, and colony forming units (CFU) were counted on the next day.”

  1. Some of your figures’ quality is poor and hardly readable (specially fig 2, fig 4, also oethers). Please revise and prepare with much higher resolution.

The authors’ answer:

Thanks for the heads up. In the revision, besides improving the resolution of Figures 2 and 4, we also provided clearer versions of other figures. 

  1. Lines 274- 286, mostly are methods… it should be appeared in the methods… may be a brief and might be appeared in the result section. Please carefully elaborate and appropriately remove form here.

The authors’ answer:

Thanks for your suggestion. In the revised manuscript, we moved these contents to the methods and abbreviated this section as follows, “Combining Alphafold2 and IEDB predicted T-cell epitopes and B-cell epitopes in nine candidate proteins. Finally, we identified 16 peptides that contained more than two types of epitopes at the same time.” 

  1. Lines 365-366, this is not result. Please move to the introduction part.

The authors’ answer:

Thank you for the friendly reminder. We also agree that this sentence should not appear in the “Results”, and for readability, the former Lines 365-366 was revised as: “To confirm rMVSS could induce an immune response, in silico immune simulation was performed, and the results were shown the antibody titer significantly increased with second boost vaccination in Figure S5 (Supplementary Information).”

  1. It is unclear how much blood was taken? How about the approach of taking blood? Please elaborate in the text. In fig 5D it is written orbital bleeding? Why not other appropriate approach of taking blood samples for suitablyacceptable animal welfare committee. I would not use this approach!

The authors’ answer:

Thanks for your comments. In the revised version, we elaborated the above-mentioned details in the manuscript:

“To minimize pain and mortality in mice, the orbital blood collection procedure strictly complied with the guidelines of the National Centre for Replacement, Refinement and Reduction of Animals in Research (NC3Rs) and published articles[5,6]. Briefly, 200 µL of whole blood was collected from the retro-orbital sinus of anesthetized mice (n=3) using non-heparinized capillary puncture and immediately transferred to a 1.5 mL sterile centrifuge tube at 37°C for 1 h followed by overnight at 4°C to separate the serum. After blood collection, the capillary was gently pulled out, the eyelids were closed, and a cotton pad was lightly pressed to minimize bleeding. During blood collection, The mice did not show any sign of consciousness during the sampling procedure.”

We strongly agree with you that the orbital blood collection is a relatively rude practice, and we will prefer saphenous vein puncture that maximize guaranteed animal welfare in future animal experiments

  1. About Fig 5,its caption, your should clearly mention the expression of what?Also marker (M) normally used for proteins… but for 5A you should define and replace “M” by “ladder”. Please elaborate here and throughout the text.

The authors’ answer:

Thanks for your careful checks. In the revised version, we emphasized the name of the expressed protein, described as “The recombinant plasmid pET-28a-MVSS was introduced into the expression bacterium BL21 (Figure 5A), and rMVSS was purified by His-Trap nickel column after induced expression.” In addition, we modified the caption of Figure 5 as “Recombinant protein MVSS expression, purification and immunization,” and changed "M" to "Ladder" in 5A with the description of Figure 5A: "Ladder as indicator, lane 1: pET-28a-MVSS-BL21; lane 2: pET-28a-MVSA; lane 3: negative."

Figure 5 Recombinant proteins MVSS expression, purification and immunization

  1. Your device-produced images should be more appropriately appear in the manuscript.

The authors’ answer:

Thanks for your suggestion. Regarding device-produced images, the assistant editor also reminded me by email that we should provide unprocessed western blot images. In the revised manuscript, Figure S6 showed the unprocessed western blot images, and we increased the description of this as follows:

“In addition, unprocessed-produced images showing all the bands on the Western blot in Figure S6.”

Figure S6 Western blot analyses of MVSS. M indicated Marker, lane 1 showed the reactivity of anti-rMVSS serum with rMVSS.

  1. About the numeration of mice in your in vivo study, the “n” is unclear … for survival rate in relation to “Evaluation of MVSS active immunization protection on mice infected with SSChz strain and others”number of mice? etc…? Results related to the figs 6 and 7 and others? It should be clearly and appropriately mentioned in the text and captions. Please also add much better quality for fig 8E (pathological ones).

The authors’ answer:

Thank you for the friendly reminder. In the revised manuscript, we clarified the number of mice involved in the material methods, results and figure legends, for example:

“The results showed the survival rate was 40% (4/10 mice) in the sham-immunized group and 70% (7/10 mice) in rMVSS-immunized group with 50% protection rate (Figure 6A).”

Also, we added much better quality for Fig 8E in the revision.

Reference

  • Wang, Z.; Guo, M.; Kong, L.; Gao, Y.; Ma, J.; Cheng, Y.; Wang, H.; Yan, Y.; Sun, J. TLR4 Agonist Combined with Trivalent Protein JointS of Streptococcus Suis Provides Immunological Protection in Animals. Vaccines2021, 9, 184, doi:10.3390/vaccines9020184.
  • Veterinary Sciences | Free Full-Text | Development of a Universal Multi-Epitope Vaccine Candidate against Streptococcus Suis Infections Using Immunoinformatics Approaches Available online: https://www.mdpi.com/2306-7381/10/6/383 (accessed on 11 January 2024).
  • Zhang, Y.; Liang, S.; Zhang, S.; Zhang, S.; Yu, Y.; Yao, H.; Liu, Y.; Zhang, W.; Liu, G. Development and Evaluation of a Multi-Epitope Subunit Vaccine against Group B Streptococcus Infection. Emerg. Microbes Infect. 2022, 11, 2371–2382, doi:10.1080/22221751.2022.2122585.
  • Babu L, Uppalapati SR, Sripathy MH, Reddy PN. Evaluation of Recombinant Multi-Epitope Outer Membrane Protein-Based Klebsiella pneumoniae Subunit Vaccine in Mouse Model. Front Microbiol. 2017;8:1805. doi: 10.3389/fmicb.2017.01805. PubMed PMID: 28979250.
  • Impact of Three Commonly Used Blood Sampling Techniques on the Welfare of Laboratory Mice: Taking the Animal’s Perspective | PLOS ONE Available online: https://journals.plos.org/plosone/article?id=10.1371/journal.pone.0238895 (accessed on 12 January 2024).
  • Hempel, D.M.; Smith, K.A.; Claussen, K.A.; Perricone, M.A. Analysis of Cellular Immune Responses in the Peripheral Blood of Mice Using Real-Time RT-PCR. J. Immunol. Methods 2002, 259, 129–138, doi:10.1016/S0022-1759(01)00502-6.

Reviewer 2 Report

Comments and Suggestions for Authors

In this manuscript, S. Liang and colleagues employed in silico reverse vaccinology and immunoinformatic approaches to identify 10 epitopes on six highly conserved and immunogenic surface-accessible candidate proteins in the zoonotic pathogen Streptococcus suis. The authors included these epitopes in the construction of a multi-epitope vaccine (MVSS) against S. suis infections and evaluated the ability of the potential vaccine to induce cross-protective antibody titers in a mouse model. The authors report that anti-rMVSS serum inhibited five S. suis serotypes in vitro and had protective effects against infection in immunized mice challenged with different doses of S. suis. Overall, the experimental design described in this study is comprehensive, evaluating the constructed MVSS in terms of molecular dynamics, immune response stimulation, histopathological damage, and mouse survival after passive immunization. A few minor comments are provided below.

(1) Table 1 should be re-oriented to improve readability. Also, define abbreviations used in Table 1.

(2) Figures 2A-D are unreadable in the current format and should be modified to improve readability.

(3) Please report in the text (in section 3.7) which specific pro-inflammatory cytokines showed increased production levels following immunization with rMVSS and indicate significance.

(4) In the Figure 5 legend, Lane M in the gels corresponds to molecular weight "Markers"--not "Maker" as given in the text. This should be corrected for clarity.

(5) Table S2 caption needs to be removed in its current form (p. 14, lines 434-444), and the information associated with Table S2 should be incorporated at the appropriate location in the text. 

Comments on the Quality of English Language

In general, the writing in this manuscript was clear; however, moderate editing is needed to correct grammatical errors (e.g., run-on sentences) and syntax. 

Author Response

Dear Editor and Referees,

Thank you for giving us the opportunity to revise our manuscript.

We have revised and improved the manuscript according to the Referees′ helpful comments. Our responses to each question from the referees are attached below. We appreciate the comments and advice provided by the Editors/Reviewers and hope that the corrections will meet with your approval.
We look forward to reading your appraisal of the revised paper.

Yours sincerely,
Guangjin Liu

Replies to Referee 2

Comments:

In this manuscript, S. Liang and colleagues employed in silico reverse vaccinology and immunoinformatic approaches to identify 10 epitopes on six highly conserved and immunogenic surface-accessible candidate proteins in the zoonotic pathogenStreptococcus suis. The authors included these epitopes in the construction of a multi-epitope vaccine (MVSS) againstS. suisinfections and evaluated the ability of the potential vaccine to induce cross-protective antibody titers in a mouse model. The authors report that anti-rMVSS serum inhibited fiveS. suisserotypesin vitroand had protective effects against infection in immunized mice challenged with different doses ofS. suis. Overall, the experimental design described in this study is comprehensive, evaluating the constructed MVSS in terms of molecular dynamics, immune response stimulation, histopathological damage, and mouse survival after passive immunization. A few minor comments are provided below.

  • Table 1 should be re-oriented to improve readability. Also, define abbreviations used in Table 1.

The authors’ answer:

Thank you for your suggestion. In the revised version, we improved the readability of Table 1 and added the full names of abbreviations in Table 1, for instance, extracellular protein factor (EF), muramidase-released protein (MRP).

Table 1. Ten epitopes were chosen using immunoinformatics from six candidate proteins.

GenBank

Annotation

Selected epitope

Antigenicity score

Hydropathicity

ACS66679.1

Enolase

AKEAGYTAVVSHRSGETEDS

1.2969

-0.885

GEHEAVELRDGDKSRYLGLG

0.9193

-0.995

WP_033875493.1

Cell wall protein

GDTAGTTTDTKTPEKANDGG

2.273

-1.455

EKGVNAIVVLAHVPATSKDG

0.9026

0.255

WP_004298861.1

Penicillin-binding protein 2B

LNILFSIVIFLFLVLILRLA

2.0369

2.72

NGPRTEINMKKRKNKPLEHD

0.9815

-2.145

WP_004195559.1

hypothetical protein

GEEEHEGHDHSEEGHSHAYD

1.7881

-2.315

CAA50714.1

extracellular protein factor (EF)

IAGYRTVNSDGTKTETVEET

1.5104

-0.88

ACN96609.1

muramidase-released protein (MRP)

TTPGTNGEVPNIPYVPGYTP

1.1127

-0.61

TKDGLRYVLVPSKTDGEENG

1.0219

-1.005

  • Figures 2A-D are unreadable in the current format and should be modified to improve readability.

The authors’ answer:

Thanks for the heads up. In the revised version, we altered the layout of Figure 2 and provided a high-resolution format for readability.

Figure 2 Pan-genomic analysis of 120 S. suis strains.

  • Please report in the text (in section 3.7) which specific pro-inflammatory cytokines showed increased production levels following immunization with rMVSS and indicate significance.

The authors’ answer:

Thanks for your comments. In the revised manuscript, we described which specific pro-inflammatory cytokine production levels were increased after inoculation with rMVSS in the text (section 3.7) and indicated their significance as follows:

“Among them, IFN-γ, an indicator of Th1-type immune response, is an important assessment for the development of porcine streptococcal vaccines[1,2], and it was strongly induced in rMVSS simulated immunization. In addition IL-2, which is associated with the prevention of pathogenic bacterial infections[3], was also strongly stimulated to be upregulated with multiple immunizations against MVSS.”

(4) In the Figure 5 legend, Lane M in the gels corresponds to molecular weight "Markers"--not "Maker" as given in the text. This should be corrected for clarity.

The authors’ answer:

Thanks for your careful checks. In the revised manuscript, we corrected "Maker" to "Marker" .

(5)Table S2caption needs to be removed in its current form (p. 14, lines 434-444), and the information associated with Table S2 should be incorporated at the appropriate location in the text.

The authors’ answer:

Thanks for your careful checks. We also do not know why "Table S2. strain D74-2 (3×108 cfu/mouse)" appears here. In fact, our manuscript did not have Table S2. So In the revised manuscript, we have revised this section:“To evaluate the cross-protective of MVSS vaccine against S. suis infections, triple immunized mice were similarly infected with SS2 strain D74-2 (3×108 cfu/mouse).” 

Reference

  • Yi L, Du Y, Mao C, Li J, Jin M, Sun L, Wang Y. Immunogenicity and protective ability of RpoE against Streptococcus suis serotype 2. J Appl Microbiol. 2021 Apr;130(4):1075-1083. doi: 10.1111/jam.14874. PubMed PMID: 32996241.
  • Li Q, Lv Y, Li YA, Du Y, Guo W, Chu D, Wang X, Wang S, Shi H. Live attenuated Salmonella enterica serovar Choleraesuis vector delivering a conserved surface protein enolase induces high and broad protection against Streptococcus suis serotypes 2, 7, and 9 in mice. Vaccine. 2020 Oct 14;38(44):6904-6913. pii: S0264-410X(20)31114-2. doi: 10.1016/j.vaccine.2020.08.062. PubMed PMID: 32907758.
  • Rubins JB, Pomeroy C. Role of gamma interferon in the pathogenesis of bacteremic pneumococcal pneumonia. Infect Immun. 1997 Jul;65(7):2975-7. doi: 10.1128/iai.65.7.2975-2977.1997. PubMed PMID: 9199475.

Round 2

Reviewer 1 Report

Comments and Suggestions for Authors

-Thanks for your effort, but it needs some more efforts to be acceptable.

-I would never use orbital-sinus for such blood collection….

About your qPCR cycle are you sure on the point mentioned (by 40 cycles of 95°C for 10s, 60°C for 30s, and finally 95°C for 30s, 60°C for 30s, and 95°C for 15s)…. After annealing you used 95°C for 30s during 40 times?? And again 60°C for 30s40 times? Please check and correct this part?

- Fig 2 even with highest magnification unreadable…., quality and readability of the Fig 2 should be improved?

Table for primers should be presented with a more informative manner. Please elaborate.

Good luck

Comments on the Quality of English Language

Improvement needed.

Author Response

Dear Editor and Referees,

Thank you for your e-mail together with the reviewer's kind comments concerning our manuscript entitled “Combined immunoinformatics to design and evaluate a multi-epitope vaccine candidate against Streptococcus suis infection”.

We have revised the manuscript according to the reviewer's comments. When the text was changed, it is highlighted by bule-coloring in the revised manuscript for easier tracking. Enclosed below are the revised points including point-to-point response to the comments.

We hope the revised manuscript could be acceptable for publication in “Vaccines”. Thank you in advance for your time and effort! We look forward to reading your appraisal of the revised paper.

Sincerely yours,

Guangjin Liu

Thanks for your effort, but it needs some more efforts to be acceptable.

  1. I would never use orbital-sinus for such blood collection….

The authors’ answer:

Thanks for your comment. We strongly agree with you that the orbital blood collection is a relatively rude practice. To minimize pain, we referred to the guidelines of NC3Rs and the protocols in the published literature[1,2], as well as the mice were anesthetized throughout the blood collection process without any signs of sentience. Considering the principle of “Reduction” and your comment, the blood collection from orbital-sinus would never be used in our lab. In future animal experiments, we will prioritize methods that maximize animal welfare such as saphenous vein puncture.

  1. About your qPCR cycle are you sure on the point mentioned (by 40 cycles of 95°C for 10s, 60°C for 30s, and finally 95°C for 30s, 60°C for 30s, and 95°C for 15s)…. After annealing you used 95°C for 30s during 40 times?? And again 60°C for 30s40 times? Please check and correct this part?

The authors’ answer:

Thanks for your careful checks. The qPCR reaction parameters were fully referenced to the standard program in the qPCR Master Mix kit (China Vazyme) instruction manual. To accurately describe the reaction parameters, lines 255-256 were changed in the revised version as :“ Briefly, the first stage is an initial denaturation program of 30s at 95°C, the second stage is the 40 cycling reactions of 10s at 95°C and 30s at 60°C, and the final stage is melting curve of 30s at 95°C, 30s at 60°C and 15s at 95°C.”About the instruction manual, if you need more informations, you can look up the website below:

" https://www.vazymebiotech.com/uploads/Vazyme-Q711_Manual-V22.1.pdf ".

  1. Fig 2 even with highest magnification unreadable…., quality and readability of the Fig 2 should be improved?

The authors’ answer:

Thanks for your kind reminder. In the revised version, we provide the highest quality of the Fig. 2 within the range allowed by the journal.

Figure 2 Pan-genomic analysis of 120 S. suis strains.

  1. Table for primers should be presented with a more informative manner. Please elaborate.

The authors’ answer:

Thanks for your suggestions. In the revised version, we detailed the functions and references corresponding to each primer pair in table S1 as follows;

Table S1 Primers used in this study

Primers

Sequence5’-3’

Function

Reference

MVSS-F

GGATCCGGGCCAGGTCCCGGAT

A fragment for recombinant protein MVSS gene

This study

MVSS-R

GAGCTCGAGTCGTGTTCCAGCG

T7-F

TAATACGACTCACTATAGGG

A fragment for plasmid pET-28a gene

[3]

T7-R

TGCTAGTTATTGCTCAGCGG

GAPDH-F

TGCACCACCAACTGCTTAG

A fragment for murine housekeeping marker gene

GAPDH-R

GGATGCAGGGATGATGTTC

IL-1β-F

TCCAGGATGAGGACATGAGCAC

A fragment for murine IL-1β gene

IL-1β-R

GAACGTCACACACCAGCAGGTTA

IL-6-F

CCACTTCACAAGTCGGAGGCTTA

A fragment for murine IL-6 gene

IL-6-R

GCAAGTGCATCATCGTTGTTCATAC

TNFα-F

GGTGCCTATGTCTCAGCCTCTT

A fragment for murine TNFα gene

TNFα-R

GCCATAGAACTGATGAGAGGGAG

IL-2-F

GCGGCATGTTCTGGATTTGACTC

A fragment for murine IL-2 gene

IL-2-R

CCACCACAGTTGCTGACTCATC

IL-10-F

CGGGAAGACAATAACTGCACCC

A fragment for murine IL-10 gene

IL-10-R

CGGTTAGCAGTATGTTGTCCAGC

Reference

  • Impact of Three Commonly Used Blood Sampling Techniques on the Welfare of Laboratory Mice: Taking the Animal’s Perspective | PLOS ONE Available online: https://journals.plos.org/plosone/article?id=10.1371/journal.pone.0238895 (accessed on 12 January 2024).
  • Hempel, D.M.; Smith, K.A.; Claussen, K.A.; Perricone, M.A. Analysis of Cellular Immune Responses in the Peripheral Blood of Mice Using Real-Time RT-PCR. J. Immunol. Methods 2002, 259, 129–138, doi:10.1016/S0022-1759(01)00502-6.
  • Zhang, Y.; Liang, S.; Zhang, S.; Zhang, S.; Yu, Y.; Yao, H.; Liu, Y.; Zhang, W.; Liu, G. Development and Evaluation of a Multi-Epitope Subunit Vaccine against Group B Streptococcus Infection. Emerg. Microbes Infect. 2022, 11, 2371–2382, doi:10.1080/22221751.2022.2122585.